# HAP: Structure-Aware Masked Image Modeling for Human-Centric Perception

**Junkun Yuan**[1,2][*], **Xinyu Zhang**[2][*‡], **Hao Zhou**[2], **Jian Wang**[2], **Zhongwei Qiu**[3], **Zhiyin Shao**[4], **Shaofeng Zhang**[5], **Sifan Long**[6], **Kun Kuang**[1‡] , **Kun Yao**[2], **Junyu Han**[2], **Errui Ding**[2], **Lanfen Lin**[1], **Fei Wu**[1], **Jingdong Wang**[2‡]

[1]Zhejiang University    [2]Baidu VIS    [3]University of Science and Technology Beijing
[4]South China University of Technology    [5]Shanghai Jiao Tong University    [6]Jilin University
{zhangxinyu14,wangjingdong}@baidu.com    {yuanjk,kunkuang}@zju.edu.cn
Code: https://github.com/junkunyuan/HAP
Project Page: https://zhangxinyu-xyz.github.io/hap.github.io/

## Abstract

Model pre-training is essential in human-centric perception. In this paper, we first introduce masked image modeling (MIM) as a pre-training approach for this task. Upon revisiting the MIM training strategy, we reveal that human structure priors offer significant potential. Motivated by this insight, we further incorporate an intuitive human structure prior - human parts - into pre-training. Specifically, we employ this prior to guide the mask sampling process. Image patches, corresponding to human part regions, have high priority to be masked out. This encourages the model to concentrate more on body structure information during pre-training, yielding substantial benefits across a range of human-centric perception tasks. To further capture human characteristics, we propose a structure-invariant alignment loss that enforces different masked views, guided by the human part prior, to be closely aligned for the same image. We term the entire method as HAP. HAP simply uses a plain ViT as the encoder yet establishes new state-of-the-art performance on 11 human-centric benchmarks, and on-par result on one dataset. For example, HAP achieves 78.1% mAP on MSMT17 for person re-identification, 86.54% mA on PA-100K for pedestrian attribute recognition, 78.2% AP on MS COCO for 2D pose estimation, and 56.0 PA-MPJPE on 3DPW for 3D pose and shape estimation.

## 1 Introduction

Human-centric perception has emerged as a significant area of focus due to its vital role in real-world applications. It encompasses a broad range of human-related tasks, including person re-identification (ReID) [23, 32, 62, 90], 2D/3D human pose estimation [15, 78, 82], pedestrian attribute recognition [34, 36, 63], *etc*. Considering their training efficiency and limited performance, recent research efforts [8, 13, 16, 33, 64] have dedicated to developing general human-centric pre-trained models by using large-scale person datasets, serving as foundations for diverse human-centric tasks.

Among these studies, some works [8, 13, 33] employ contrastive learning for self-supervised pre-training, which reduces annotation costs of massive data in supervised pre-training [16, 64]. In this work, we take the first step to introduce masked image modeling (MIM), which is another form of self-supervised learning, into human-centric perception, since MIM [2, 9, 30, 77] has shown considerable potential as a pre-training scheme by yielding promising performance on vision tasks [44, 65, 78].

---

[*]Equal contribution. This work was done when Junkun Yuan, Zhongwei Qiu, Zhiyin Shao, and Shaofeng Zhang were interns at Baidu VIS.    [‡]Corresponding author.

37th Conference on Neural Information Processing Systems (NeurIPS 2023).

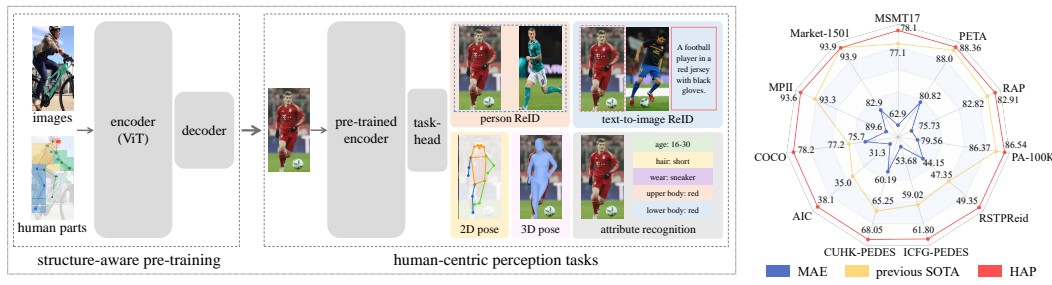

| (a) Human-centric perception with HAP. | (b) High performance of HAP. |

Figure 1: (a) Overview of our proposed HAP framework. HAP first utilizes human structure priors, *e.g.*, human body parts, for human-centric pre-training. HAP then applies the pre-trained encoder, along with a task-specific head, for addressing each of the human-centric perception tasks. (b) Our HAP demonstrates state-of-the-art performance across a broad range of human-centric benchmarks.

There is no such thing as a free lunch. Upon initiating with a simple trial, *i.e.*, directly applying the popular MIM method [30] with the default settings for human-centric pre-training, it is unsurprising to observe poor performance. This motivates us to delve deep into understanding why current MIM methods are less effective in this task. After revisiting the training strategies, we identify three crucial factors that are closely related to human structure priors, including high scaling ratio[2], block-wise mask sampling strategy, and intermediate masking ratio, have positive impacts on human-centric pre-training. Inspired by this, we further tap the potential of human structure-aware prior knowledge.

In this paper, we incorporate human parts [5, 26, 57, 60, 80], a strong and intuitive human structure prior, into human-centric pre-training. This prior is employed to guide the mask sampling strategy. Specifically, we randomly select several human parts and mask the corresponding image patches located within these selected regions. These part-aware masked patches are then reconstructed based on the clues of the remaining visible patches, which can provide semantically rich body structure information. This approach encourages the model to learn contextual correlations among body parts, thereby enhancing the learning of the overall human body structure. Moreover, we propose a structure-invariant alignment loss to better capture human characteristics. For a given input image, we generate two random views with different human parts being masked out through the proposed mask sampling strategy. We then align the latent representations of the two random views within the same feature space. It preserves the unique structural information of the human body, which plays an important role in improving the discriminative ability of the pre-trained model for downstream tasks.

We term the overall method as HAP, short for **H**uman structure-**A**ware **P**re-training with MIM for human-centric perception. HAP is simple in terms of modalities, data, and model structure. We unify the learning of two modalities (images and human keypoints) within a single dataset (LUPerson [23]). The model structure is based on the existing popular MIM methods, e.g., MAE [30] and CAE [9], with a plain ViT [21] as the encoder, making it easy to transfer to a broad range of downstream tasks (see Figure 1a). Specifically, HAP achieves state-of-the-art results on 11 human-centric benchmarks (see Figure 1b), *e.g.*, 78.1% mAP on MSMT17 for person ReID, 68.05% Rank-1 on CUHK-PEDES for text-to-image person ReID, 86.54% mA on PA-100K for pedestrian attribute recognition, and 78.2% AP on MS COCO for 2D human pose estimation. In summary, our main contributions are:

- We are the first to introduce MIM as a human-centric pre-training method, and reveal that integrating the human structure priors are beneficial for MIM in human-centric perception.

- We present HAP, a novel method that incorporates human part prior into the guidance of the mask sampling process to better learn human structure information, and design a structure-invariant alignment loss to further capture the discriminative characteristics of human body structure.

- The proposed HAP is simple yet effective that achieves superior performance on 11 prominent human benchmarks across 5 representative human-centric perception tasks.

---

[2]We simply refer the lower bound for the ratio of random area to crop before resizing as "scaling ratio". Please find more discussions in Section 3

## 2  Related work

**Human-centric perception** includes a broad range of human-related tasks. The goal of person Re-ID [24, 32, 51, 50, 81, 90, 91, 84, 86] and text-to-image person ReID [59, 20, 62, 58] is to identify persons of interest based on their overall appearance or behavior. Pedestrian attribute recognition [35, 36] aims to classify the fine-grained attributes and characteristics of persons. 2D [78, 79, 82] and 3D [69, 14, 25] pose estimation learn to predict position and orientation of individual body parts.

Recent works make some efforts to pre-train a human-centric model. HCMoCo [33] learns modal-invariant representations by contrasting features of dense and sparse modalities. LiftedCL [13] extracts 3D information by adversarial learning. UniHCP [16] and PATH [64] design multi-task co-training approaches. SOLIDER [8] tries to balance the learning of semantics and appearances.

Compared with these works, this paper focuses on exploring the advantage of utilizing human structure priors in human-centric pre-training. Besides, instead of training with numerous modalities and datasets using a specially designed structure, we pursue a simple pre-training framework by employing two modalities (images and human keypoints) of one dataset (LUPerson [23]) and vanilla ViT structure with the existing popular MIM methods (*e.g.*, BEiT [2], MAE [30], and CAE [9]).

**Self-supervised learning** has two main branches: contrastive learning (CL) and masked image modeling (MIM). CL [4, 7, 10, 11, 12, 28, 31] is a discriminative approach that learns representations by distinguishing between similar and dissimilar image pairs. Meanwhile, MIM [2, 9, 30, 56, 72, 73, 77, 85, 88] is a generative approach that learns to recover the masked content in the corrupted input images. It is shown that CL learns more discriminative representations [4, 12], while MIM learns finer grained semantics [2, 30]. To obtain transferable representations for the diverse tasks, our work combines both of them to extract discriminative and fine-grained human structure information.

Semantic-guided masking is presented in some recent MIM works [6, 29, 37, 40, 45, 68, 83]. Many methods [83, 29, 6, 37] utilize the attention map to identify and mask informative image patches, which are then reconstructed to facilitate representation learning. In comparison, this paper explores to directly employ the human structure priors to guide mask sampling for human-centric perception.

## 3  HAP

Inspired by the promising performance of MIM in self-supervised learning, we initially employ MIM as the pre-training scheme for human-centric perception. Compared with MIM, our proposed HAP integrates human structure priors into MIM for human-centric pre-training, and introduces a structure-invariant alignment loss. The overview of our HAP framework is shown in Figure 2.

### 3.1  Preliminary: great potential of human structure priors

We use MAE [30] with its default settings as an example for human-centric pre-training. LUPerson [23] is used as the pre-training dataset following [8, 23, 51]. We designate this setting as our *baseline*. The results are shown in Figure 3. The baseline achieves poor performance on two representative human-centric perception tasks, i.e., person ReID and 2D human pose estimation. This phenomenon motivates us to delve deep into understanding the underlying reasons.

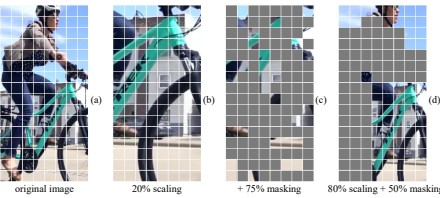

Figure 4: **Visualization of training strategies.** For a given image (a), the baseline [30] uses 20% scaling ratio (b) and 75% masking ratio (c) with random mask sampling strategy, yielding a meaningless image with little human structure information. We adopt 80% scaling ratio and 50% masking ratio with block-wise mask sampling (d), maintaining the overall body structure.

We start by visualizing the input images under different training strategies in Figure 4. Surprisingly, we observe that it is hard to learn information of human body structure from the input images of the previous baseline (see Figure 4 (b-c)). We conjecture that: i) the scaling ratio may be too small to preserve the overall human body structure, as shown in Figure 4 (b), and ii) the masking ratio may be too large to preserve sufficient human structure information, which makes the reconstruction task too difficult to address, as shown in Figure 4 (c).

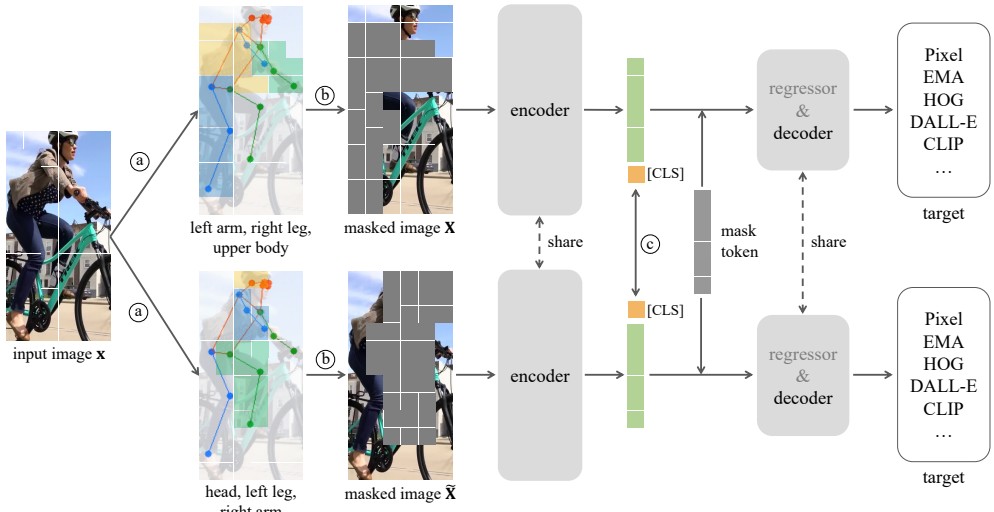

(a) randomly select human parts for masking  (b) adjust masked patch number to match the pre-defined masking ratio  (c) structure-invariant alignment

Figure 2: **Our HAP framework.** During pre-training, we randomly select human part regions and mask the corresponding image patches. We then adjust the masked patches to match the pre-defined masking ratio. The encoder embeds the visible patches, and the decoder reconstructs the masked patches based on latent representations. For a given image, HAP generates two views using the mask sampling with human part prior. We apply the proposed structure-invariant alignment loss to bring the [CLS] token representations of these two views closer together. After pre-training, only the encoder with a plain ViT structure is retained for transferring to downstream human-centric perception tasks.

Driven by this conjecture, we further study the impacts of scaling ratio and masking ratio in depth. Figure 3 illustrates that both *i) high scaling ratio* (ranging from 60% to 90%) and *ii) mediate masking ratio* (ranging from 40% to 60%) contribute positively to the performance improvement of human-centric tasks. Especially, an 80% scaling ratio can bring an approximately 3.5% improvement on MSMT17 of the person ReID task. This effectively confirms that information of human body structure is crucial during human-centric pre-training. Additionally, we also discover that *iii) block-wise mask sampling strategy* performs sightly better than the random mask sampling strategy, since the former one masks semantic human bodies appropriately due to block regions. We refer to these three factors of the training strategy as *human structure priors* since they are associated with human structure information. In the following exploration, we employ scaling ratio of 80%, masking ratio of 50% with block-wise masking strategy by default. See an example of our optimal strategy in Figure 4 (d).

## 3.2   Human part prior

Inspired by the above analyses, we consider to incorporate human body parts, which are intuitive human structure priors, into human-centric pre-training. Here, we use 2D human pose keypoints [78], which are convenient to obtain, to partition human part regions by following previous works [60]. Section 3.1 reveals that the mask sampling strategy plays an important role in human-centric pre-training. To this end, we directly use human structure information provided by human body parts as robust supervision to guide the mask sampling process of our HAP during human-centric pre-training.

Specifically, we divide human body into six parts, including head, upper body, left/right arm, and left/right leg according to the extracted human keypoints, following [60]. Given a specific input image $x$, HAP first embeds $x$ into $N$ patches. With a pre-defined masking ratio $\beta$, the aim of mask sampling is to mask $N_m = \max\{n \in \mathcal{Z} | n \leq \beta \times N\}$ patches, where $\mathcal{Z}$ represents the set of integers. Specifically, we randomly select $P$ parts, where $0 \leq P \leq 6$. Total $N_p$ image patches located on these $P$ part regions are masked out. Then we adjust the number of masked patches to $N_m$ based on the instructions introduced later. In this way, unmasked (visible) human body part regions provide semantically rich human structure information for the reconstruction of the masked human body parts, encouraging the pre-trained model to capture the latent correlations among the body parts.

There are three situations for the masked patches adjustment: i) if $N_p = N_m$, everything is well-balanced; ii) if $N_p < N_m$, we further use block-wise sampling strategy to randomly select additional

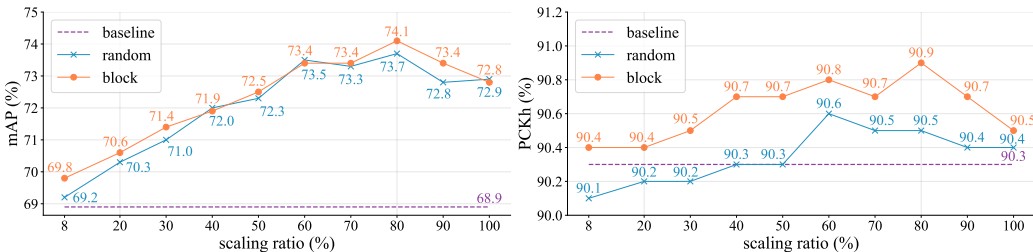

(a) **Scaling ratio** analysis on (left) MSMT17 and (right) MPII. A scaling ratio with 60%-90% works well.

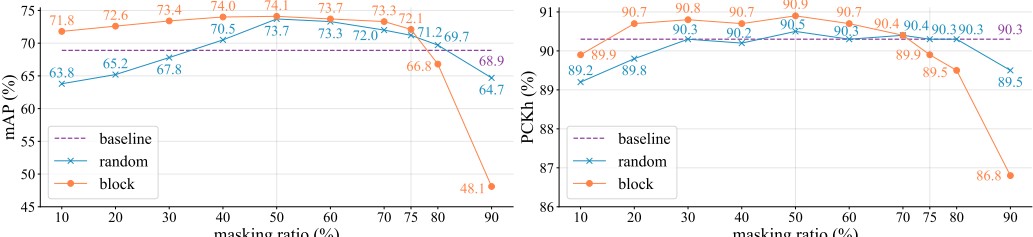

(b) **Masking ratio** analysis on MSMT17 (left) and MPII (right). A masking ratio with 40%-60% works well.

Figure 3: **Study on training strategy.** We investigate three factors of training strategy, *i.e.*, (a) scaling ratio, (b) masking ratio, and (a,b) mask sampling strategy, on MSMT17 [74] (person ReID) and MPII [1] (2D pose estimation). The baseline is MAE [30] with default settings (20% scaling ratio, 75% masking ratio, random mask sampling strategy). The optimal training strategy for human-centric pre-training is with 80% scaling ratio, 50% masking ratio, and block-wise mask sampling strategy.

$N_m - N_p$ patches; iii) if $N_p > N_m$, we delete patches according to the sequence of human part selection. We maintain a total of $N_m$ patches, which contain structure information, to be masked.

**Discussion.** i) We use human parts instead of individual keypoints as guidance for mask sampling. This is based on the experimental result in Section 3.1, *i.e.*, the block-wise mask sampling strategy outperforms the random one. More experimental comparisons are available in Appendix. ii) HCMoCo [33] also utilizes keypoints during human-centric pre-training, in which sparse keypoints serve as one of the multi-modality inputs. Different from HCMoCo, we integrate two modalities, including images and keypoints, together for mask sampling. It should be noted that the keypoints in our method are pseudo labels; therefore they may not be accurate enough for supervised learning as in HCMoCo. However, these generated keypoints can be beneficial for the self-supervised learning.

## 3.3 Structure-invariant alignment

Learning discriminative human characteristics is essential in human-centric perception tasks. For example, person ReID requires to distinguish different pedestrians based on subtle details and identify the same person despite large variances. To address this issue, we further propose a structure-invariant alignment loss, to enhance the representation ability of human structural semantic information.

For each person image $\mathbf{x}$, we generate two different views through the proposed mask sampling strategy with guidance of human part prior. These two views, denoted as $\mathbf{X}$ and $\widetilde{\mathbf{X}}$, display different human part regions for the same person. After model encoding, each view obtains respective latent feature representations $\mathbf{Z}$ and $\widetilde{\mathbf{Z}}$ on the normalized [CLS] tokens. Our structure-invariant alignment loss $\mathcal{L}_{\text{align}}$ is applied to $\mathbf{Z}$ and $\widetilde{\mathbf{Z}}$ for aligning them. We achieve it by using the InfoNCE loss [54]:

$$\mathcal{L}_{\text{align}} = -\log \frac{\exp(\mathbf{Z} \cdot \widetilde{\mathbf{Z}}/\tau)}{\sum_{i=1}^{B} \exp(\mathbf{Z} \cdot \mathbf{Z}_i/\tau)}, \tag{1}$$

where $B$ is the batch size. $\tau$ is a temperature hyper-parameter set to 0.2 [12]. In this way, the encoder is encouraged to learn the invariance of human structure, and its discriminative ability is improved.

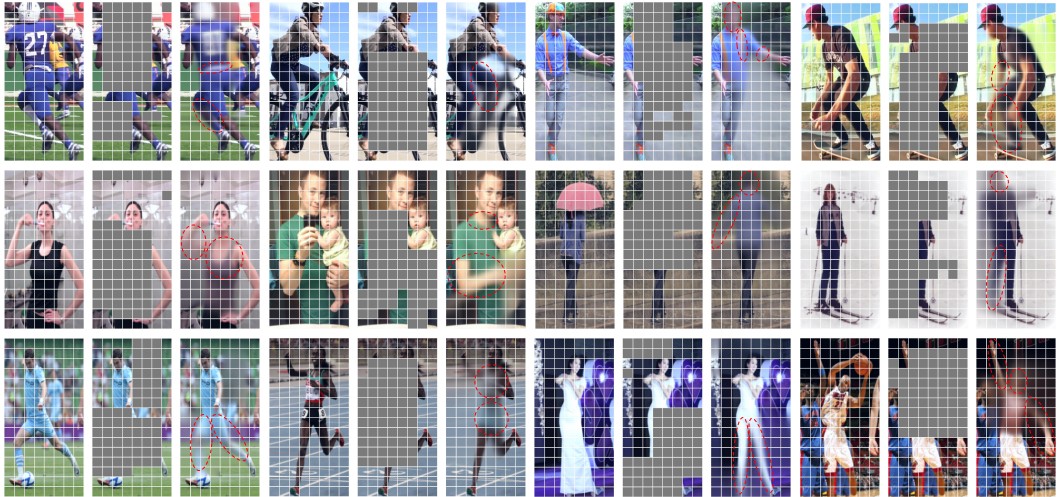

Figure 5: **Reconstructions** of LUPerson (1st row), COCO (2nd row), and AIC (3rd row) images using our HAP pre-trained model. The pre-training dataset is LUPerson. For each triplet, we show (left) the original image, (middle) the masked image, and (right) images reconstructed by HAP. We highlight some interesting observations using red dotted ellipses. Although the reconstructed parts are not exactly identical to the original images, they are semantically reasonable as body parts. For example, in the first triplet of the 1st row, the reconstructed right leg of the player differs from the original, but still represents a sensible leg. It clearly shows that HAP effectively learns the semantics of human body structure and generalizes well across various domains. Zoom in for a detailed view.

### 3.4 Overall: HAP

To sum up, in order to learn the fine-grained semantic structure information, our HAP incorporates the human part prior into the human-centric pre-training, and introduces a structure-invariant alignment loss to capture discriminative human characteristics. The whole loss function to be optimized is:

$$\mathcal{L} = \mathcal{L}_{\text{recon}} + \gamma \mathcal{L}_{\text{align}}, \tag{2}$$

where $\mathcal{L}_{\text{recon}}$ is the MSE loss for reconstruction following [30]. $\gamma$ is the weight to balance the two loss functions of $\mathcal{L}_{\text{recon}}$ and $\mathcal{L}_{\text{align}}$. We set $\gamma$ to 0.05 by default, and leave its discussion in Appendix.

## 4 Experiments

### 4.1 Settings

**Pre-training.** LUPerson [23] is a large-scale person dataset, consisting of about 4.2M images of over 200K persons across different environments. Following [51], we use the subset of LUPerson with 2.1M images for pre-training. The resolution of the input image is set to $256 \times 128$ and the batch size is set to 4096. The human keypoints of LUPerson, *i.e.*, the human structure prior of HAP, are extracted by ViTPose [78]. The encoder model structure of HAP is based on the ViT-Base [21]. HAP adopts AdamW [49] as the optimizer in which the weight decay is set to 0.05. We use cosine decay learning rate schedule [48], and the base learning rate is set to 1.5e-4. The warmup epochs are set to 40 and the total epochs are set to 400. The model of our HAP is initialized from the official MAE/CAE model pre-trained on ImageNet by following the previous works [16, 78]. After pre-training, the encoder is reserved, along with task-specific heads, for addressing the following human-centric perception tasks, while other modules (*e.g.*, regressor and decoder) are discarded.

**Human-centric perception tasks.** We evaluate our HAP on 12 benchmarks across 5 human-centric perception tasks, including *person ReID* on Market-1501 [87] and MSMT17 [74], *2D pose estimation* on MPII [1], COCO [46] and AIC [75], *text-to-image person ReID* on CUHK-PEDES [42], ICFG-PEDES [20] and RSTPReid [89], *3D pose and shape estimation* on 3DPW [67], *pedestrian attribute*

Table 1: **Main results.** We compare HAP with representative task-specific methods, human-centric pre-training methods, and the baseline MAE [30]. ‡ indicates using stronger model structures such as sliding windows in [32, 8] and convolutions in [50]. HAP‡ follows the implementation of [50].

(a) **Statistics** on the number of training datasets and training data samples used during the pre-training process of the human-centric pre-training methods.

| method | datasets | samples |
|---|---|---|
| LiftedCL [13] | 1 | ∼150K |
| SOLIDER [8] | 1 | ∼4.2M |
| HCMoCo [33] | 2 | ∼82K |
| UniHCP [16] | 33 | ∼2.3M |
| PATH [64] | 37 | ∼11.0M |
| HAP | 1 | ∼2.1M |

(b) **Attribute recognition.** mA (%) is reported.

| method | PA-100K | RAP | PETA |
|---|---|---|---|
| ALM [63] | 80.68 | 81.87 | 86.30 |
| DAFL [35] | 83.54 | 81.04 | 87.07 |
| RethinkPAR [36] | 81.61 | 80.82 | 85.17 |
| Label2Label [43] | 82.24 | - | - |
| PATH [64] | 85.0 | 81.2 | 88.0 |
| UniHCP [16] | 86.18 | 82.34 | - |
| SOLIDER‡ [8] | 86.37 | - | - |
| MAE [30] | 79.56 | 75.73 | 80.82 |
| HAP | **86.54** | **82.91** | **88.36** |

(c) **Person ReID**. mAP(%) is reported. For fair comparisons, we present all the results according to the input image resolution of $256 \times 128$ (256) and $384 \times 128$ (384) respectively for each compared method.

| method | MSMT17 | | Market-1501 | |
|---|---|---|---|---|
| resolution | 256 | 384 | 256 | 384 |
| TransReID [32] | 64.9 | 66.6 | 88.2 | 88.8 |
| TransReID‡ [32] | 67.4 | 69.4 | 88.9 | 89.5 |
| UP-ReID [81] | 63.3 | - | 91.1 | - |
| LUP [23] | 65.7 | - | 91.0 | - |
| PASS [90] | 71.8 | 74.3 | 93.0 | 93.3 |
| MALE‡ [50] | 73.0 | 74.2 | 92.2 | 92.1 |
| TransReID-SSL‡ [51] | - | 75.0 | - | 93.2 |
| PATH [64] | 69.1 | - | 89.5 | - |
| UniHCP [16] | 67.3 | - | 90.3 | - |
| SOLIDER‡ [8] | - | 77.1 | - | **93.9** |
| MAE [30] | 62.0 | 62.9 | 82.9 | 82.5 |
| HAP | 76.4 | 76.8 | 91.7 | 91.9 |
| HAP‡ | **78.0** | **78.1** | **93.8** | 93.9 |

(d) **2D pose estimation.** PCKh (%) is reported for MPII and AP (%) is reported for COCO and AIC. † represents using multi-dataset training of COCO+AIC+MPII [78]. + represents using a larger image size of $384 \times 288$.

| method | MPII | COCO | AIC |
|---|---|---|---|
| HRNet-w48 [61] | 90.1 | 75.1 | 33.5 |
| HRFormer‡ [82] | - | 75.6 | 34.4 |
| HRFormer‡+ [82] | - | 77.2 | - |
| ViTPose [78] | - | 75.8 | - |
| ViTPose† [78] | 93.3 | 77.1 | 32.0 |
| LiftedCL [13] | 89.3 | 71.1 | - |
| PATH [64] | 93.3 | 76.3 | 35.0 |
| UniHCP [16] | - | 76.5 | 33.6 |
| SOLIDER‡+ [8] | - | 76.6 | - |
| MAE [30] | 89.6 | 75.7 | 31.3 |
| HAP | 91.8 | 75.9 | 31.5 |
| HAP† | 93.4 | 77.0 | 32.2 |
| HAP+ | 92.6 | 77.2 | 37.7 |
| HAP†+ | **93.6** | **78.2** | **38.1** |

(e) **Text-to-image ReID.** Rank-1(%) is reported.

| method | CUHK-PEDES | ICFG-PEDES | RSTPReid |
|---|---|---|---|
| SAFA [41] | 64.13 | - | - |
| LBUL [71] | 61.95 | - | 43.35 |
| CAIBC [70] | 64.43 | - | 47.35 |
| SSAN [20] | 61.37 | 54.23 | - |
| SRCF [62] | 64.04 | 57.18 | - |
| LGUR [59] | 65.25 | 59.02 | - |
| MAE [30] | 60.19 | 53.68 | 44.15 |
| HAP | **68.05** | **61.80** | **49.35** |

(f) **3D pose and shape estimation.** MPJPE, PA-MPJPE, and MPVPE are reported on 3DPW.

| method | MPJPE ↓ | PA-MPJPE ↓ | MPVPE ↓ |
|---|---|---|---|
| MiHu [22] | 85.1 | 54.8 | - |
| SPIN [38] | 96.9 | 59.2 | 116.4 |
| Pose2Mesh [14] | 89.5 | 56.3 | 105.3 |
| I2L-MeshNet [52] | 93.2 | 57.7 | 110.1 |
| 3DCrowNet [15] | **81.7** | **51.5** | **98.3** |
| MAE [30] | 95.6 | 58.0 | 112.7 |
| HAP | 90.1 | 56.0 | 106.3 |

*recognition* on PA-100K [47], RAP [39] and PETA [18]. For person ReID, we use the implementation of [50] and report mAP. For 2D pose estimation, the evaluation metrics are PCKh for MPII and AP for COCO and AIC. The codebase is based on [78]. Rank-1 is reported for text-to-image person ReID with the implementation of [59]. mA is reported in the pedestrian attribute recognition task with the codebase of [36]. The evaluation metrics are MPJPE/PA-MPJPE/MPVPE for 3D pose and shape estimation. Since there is no implementation with a plain ViT structure available for this task, we modify the implementation based on [15]. More training details can be found in Appendix.

Table 2: HAP with MAE [30] and CAE [9]. The evaluation metrics are the same as those in Table 1.

| | MSMT17 | | Market-1501 | | PA-100K | MPII | COCO | RSTPReid |
|---|---|---|---|---|---|---|---|---|
| | 256 | 384 | 256 | 384 | | | | |
| MAE [30] | 76.4 | 76.8 | 91.7 | 91.9 | 86.54 | 91.8 | 75.9 | 49.35 |
| CAE [9] | 76.5 | 77.0 | 93.3 | 93.1 | 86.33 | 91.8 | 75.3 | 51.70 |

## 4.2 Main results

We compare HAP with both previous task-specific methods and human-centric pre-training methods. Our HAP is simple by using only one dataset of ∼2.1M samples, compared with existing pre-training methods (Table1a), yet achieves superior performance on various human-centric perception tasks.

In Table 1, we report the results of HAP with MAE [30] on 12 datasets across 5 representative human-centric tasks. Compared with previous task-specific methods and human-centric pre-training methods, HAP achieves state-of-the-art results on **11** datasets of **4** tasks. We discuss each task next.

**Pedestrian attribute recognition.** The goal of pedestrian attribute recognition is to assign fine-grained attributes to pedestrians, such as young or old, long or short hair, *etc*. Table 1b shows that HAP surpasses previous state-of-the-art on all of the three commonly used datasets, *i.e.*, +0.17% on PA-100K [47], +0.57% on RAP [39], and +0.36% on PETA [18]. It can be observed that our proposed HAP effectively accomplishes this fine-grained human characteristic understanding tasks.

**Person ReID.** The aim of person ReID is to retrieve a queried person across different cameras. Table 1c shows that HAP outperforms current state-of-the-art results of MSMT17 [74] by +5.0% (HAP 78.0% *vs*. MALE [50] 73.0%) and +1.0% (HAP 78.1% *vs*. SOLIDER [8] 77.1%) under the input resolution of $256 \times 128$ and $384 \times 128$, respectively. HAP also achieves competitive results on Market-1501 [87], *i.e.*, 93.9% mAP with $384 \times 128$ input size. For a fair comparison, we also report the results of HAP with extra convolution modules following [50], denoted as HAP[‡]. It brings further improvement by +1.3% and +2.0% on MSMT17 and Market-1501 with resolution of 384. Review that previous ReID pre-training methods [81, 23, 90, 51] usually adopt contrastive learning approach. These results show that our HAP with MIM is more effective on the person ReID task.

**Text-to-image person ReID.** This task uses textual descriptions to search person images of interest. We use the pre-trained model of HAP as the initialization of the vision encoder in this task. Table1e shows that HAP performs the best on the three popular text-to-image benchmarks. That is, HAP achieves 68.05%, 61.80% and 49.35% Rank-1 on CUHK-PEDES [42], ICFG-PEDES [20] and RSTPReid [89], respectively. Moreover, HAP is largely superior than MAE [30], *e.g.*, 68.05% *vs*. 60.19% (+7.86%) on CUHK-PEDES. It clearly reveals that human structure priors especially human keypoints and structure-invariant alignment indeed provide semantically rich human information, performing well for human identification and generalizing across different domains and modalities.

**2D pose estimation.** This task requires to localize anatomical keypoints of persons by understanding their structure and movement. For fair comparisons, we also use multiple datasets as in [78] and larger image size as in [82, 8] during model training. Table 1d shows that HAP outperforms previous state-of-the-art results by +0.3%, +1.0%, +3.1% on MPII [1], COCO [46], AIC [75], respectively. Meanwhile, we find both multi-dataset training and large resolution are beneficial for our HAP.

**3D pose and shape estimation.** We then lift HAP to perform 3D pose and shape estimation by recovering human mesh. Considering that there is no existing method that performs this task on a plain ViT backbone, we thus follow the implementation of [15] and just replace the backbone to a plain ViT. The results are listed in Table 1f. It is interesting to find that despite this simple implementation, our HAP achieves competitive results for the 3D pose task even HAP does not see any 3D information. These observation may suggest that HAP improves representation learning ability on human structure from the 2D keypoints information, and transfers its ability from 2D to 3D.

**CAE [9] vs MAE [30].** HAP is model-agnostic that can be integrated into different kinds of MIM methods. We also apply HAP to the CAE framework [9] for human-centric pre-training. Table 2 shows that HAP with CAE [9] achieves competitive performance on various human-centric perception tasks, especially on discriminative tasks such as person ReID that HAP with CAE is superior than HAP with MAE [30] by 1.6% on Market-1501 (with the input resolution of $256 \times 128$), and text-to-image person ReID that HAP with CAE performs better than HAP with MAE by 2.35% on RSTPReid. These observations demonstrate the versatility and scalability of our proposed HAP framework.

Table 3: **Main properties** of HAP. "HSP", "HPM", and "SIA" stand for human structure priors, human part prior for mask sampling, and structure-invariant alignment, respectively. We compare HAP with the baseline of MAE ImageNet pretrained model on the representative benchmarks.

| method | HSP | HPM | SIA | MSMT17 | MPII | CUHK-PEDES | 3DPW ↓ | PA-100K |
|---|---|---|---|---|---|---|---|---|
| baseline | | | | 68.9 | 90.3 | 66.49 | 57.4 | 83.41 |
| HAP-0 | ✓ | | | 74.1 | 90.9 | 67.69 | 56.2 | 85.50 |
| HAP-1 | ✓ | ✓ | | 75.1 | 91.1 | 67.95 | 56.2 | 85.83 |
| HAP-2 | ✓ | | ✓ | 75.6 | 91.6 | 68.00 | 56.1 | 86.38 |
| HAP | ✓ | ✓ | ✓ | **76.4** | **91.8** | **68.05** | **56.0** | **86.54** |
| MAE [30] | | | | 62.0 | 89.6 | 60.19 | 58.0 | 79.56 |

Table 4: Ablation on (left) **number of samples** and (right) **training epochs** for pre-training of HAP.

| samples | MSMT17 | MPII | | epochs | MSMT17 | MPII |
|---|---|---|---|---|---|---|
| ∼0.5M | 66.9 | 90.4 | | 100 | 72.2 | 91.3 |
| ∼1.0M | 71.9 | 91.2 | | 200 | 73.9 | 91.6 |
| ∼2.1M | 76.4 | 91.8 | | 400 | 76.4 | 91.8 |

## 4.3 Main properties

We ablate our HAP in Table 3. The analyses of main properties are then discussed in the following.

**Pre-training on human-centric data.** Our baseline, using default settings of [30] and LUPerson [23] as pre-training data, has achieved great improvement than original MAE [30], *e.g.*, +6.9% mAP on MSMT17 and +6.30% Rank-1 on CUHK-PEDES. It shows that pre-training on human-centric data benefits a lot for human-centric perception by effectively bridging the domain gap between the pre-training dataset and downstream datasets. This observation is in line with previous works [23, 51].

**Human structure priors.** We denote HAP-0 to represent human structure priors, *i.e.*, large scaling ratio and block-wise masking with intermediate masking ratio, introduced in Section 3.1. HAP-0 performs better than the baseline method, especially on MSMT17 with +5.2% improvement. It clearly reveals that human structure priors can help the model learn more human body information.

**Human part prior.** We further use human part prior to guide the mask sampling process. Compared with HAP-1 and HAP-0, part-guided mask sampling strategy brings large improvement, *e.g.*, +1.0% on MSMT17 and +0.33% on PA-100K. It clearly shows that the proposed mask sampling strategy with the guidance of human part prior has positive impact on the human-centric perception tasks.

**Structure-invariant alignment.** The structure-invariant alignment loss is applied on different masked views to align them together. Compared with HAP-2 *vs.* HAP-0 and HAP *vs.* HAP-1, this alignment loss results in +1.5% and +1.3% improvement on MSMT17, and +0.7% and +0.7% improvement on MPII, respectively. It reflects that the model successfully captures human characteristics with this loss. We provide more discussions about the alignment choices as well as the weight in Appendix.

Overall, our HAP greatly outperforms the original MAE [30] and the baseline by large margins across various human-centric perception tasks, showing significant benefits of the designs of our HAP.

## 4.4 Ablation study

**Number of samples.** We compare using different number of data samples during pre-training in Table 4 (left). Only using ∼0.5M LUPerson data already outperforms original MAE [30] pre-trained on ImageNet [17]. Meanwhile, more pre-training data brings larger improvement for our HAP.

**Training epochs.** Table 4 (right) shows that pre-training only 100 epochs already yields satisfactory performance. It suggests that HAP can effectively reduces training cost. These observations indicate that HAP, as a human-centric pre-training method, effectively reduces annotation, data, and training cost. Therefore, we believe HAP can be useful for real-world human-centric perception applications.

**Visualization.** To explore the structure information learned in the plain ViT encoder of HAP, we probe its attention map by querying the patch of a random body keypoint. See visualizations in Figure 6. Interestingly, we observe that HAP captures the relation among body parts. For example,

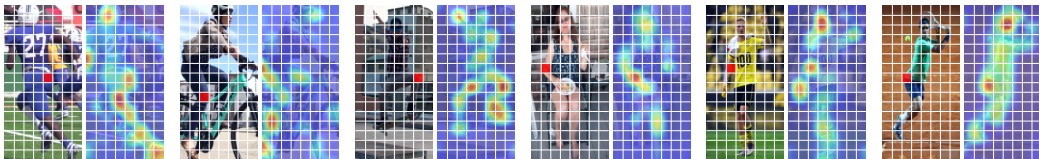

Figure 6: **Visualization on attention maps of body structure from the ViT encoder in HAP.** For each pair, we show (left) a randomly selected keypoint within a patch marked in red and (right) its corresponding attention weight to other patches learned by HAP. Patches in deeper color indicate more relevant to the selected keypoint region. The left two pairs are from LUPerson [23], the middle two are from COCO [46], and the right two are from AIC [75]. It reveals that HAP is able to learn human body structure information and successfully identifies the correlations among the body parts.

when querying the right hip of the player in the first image pair (from LUPerson), we see that there are strong relations to the left leg, left elbow, and head. And when querying the left knee of the skateboarder in the second image pair (from COCO), it is associated with almost all other keypoints. These observations intuitively illustrate that HAP learns human body structure and generalizes well.

## 5 Conclusion

This paper makes the first attempt to use Masked Image Modeling (MIM) approach for human-centric pre-training, and presents a novel method named HAP. Upon revisiting several training strategies, HAP reveals that human structure priors have great benefits. We then introduces human part prior as the guidance of mask sampling and a structure-invariant alignment loss to align two random masked views together. Experiments show that our HAP achieves competitive performance on various human-centric perception tasks. Our HAP framework is simple by unifying the learning of two modalities within a single dataset. HAP is also versatility and scalability by being applied to different MIM structures, modalities, domains, and tasks. We hope our work can be a strong baseline with generative pre-training method for the future researches on this topic. We also hope our experience in this paper can be useful for the general human-centric perception and push this frontier.

**Broader impacts.** The proposed method is pre-trained and evaluated on several person datasets, thus could reflect biases in these datasets, including those with negative impacts. When using the proposed method and the generated images in this paper, one should pay careful attention to dataset selection and pre-processing to ensure a diverse and representative sample. To avoid unfair or discriminatory outcomes, please analyze the model's behavior with respect to different demographics and consider fairness-aware loss functions or post-hoc bias mitigation techniques. We strongly advocate for responsible use, and encourage the development of useful tools to detect and mitigate the potential misuse of our proposed method and other masked image modeling techniques for unethical purposes.

**Limitations.** We use human part prior to guide pre-training, which is extracted by existing pose estimation methods. Therefore, the accuracy of these methods could affect the pre-training quality. Besides, due to the limitations in patch size, the masking of body parts may not be precise enough. If there are better ways to improve this, it should further enhance the results. Nonetheless, the proposed HAP method still shows promising performance for a wide range of human-centric perception tasks.

## Acknowledgments and Disclosure of Funding

We thank Shixiang Tang and Weihua Chen for discussions about the experiments of human-centric pre-training, and Yufei Xu and Yanzuo Lu for discussions about downstream tasks. This work was supported in part by Zhejiang Provincial Natural Science Foundation of China (LZ22F020012), National Natural Science Foundation of China (62376243, 62037001, U20A20387), Young Elite Scientists Sponsorship Program by CAST (2021QNRC001), the StarryNight Science Fund of Zhejiang University Shanghai Institute for Advanced Study (SN-ZJU-SIAS-0010), Project by Shanghai AI Laboratory (P22KS00111), Program of Zhejiang Province Science and Technology (2022C01044), the Fundamental Research Funds for the Central Universities (226-2022-00142, 226-2022-00051).

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

# A   Details of pre-training

**Model structure.** We adopt MAE-base [30] as the basic structure of HAP. It consists of an encoder and a lightweight decoder. The encoder follows the standard ViT-base [21] architecture, comprising 12 common Transformer blocks [66] with 12 attention heads and a hidden size of 768. The decoder is relatively small, which consists of 8 Transformer blocks with 16 attention heads and a hidden size of 512. Sine-cosine position embeddings are added to representations of both encoder and decoder[3]. We also use CAE-base [9] as the basic structure. Please refer to [9] for more details of the structure.

**Training data.** LUPerson [23] is a large-scale person dataset with ∼4.2M images (see Figure 8a). Following [51], we use a subset of LUPerson with ∼2.1M images for pre-training. The resolution of the input images is set to $256 \times 128$ following previous works [23, 90, 51, 50]. The data augmentation follows [50], which involves random resizing and cropping, and horizontal flipping. We utilize ViTPose [78], a popular 2D pose estimation method, to extract the keypoints of LUPerson as the human structure prior in our HAP. Given that our HAP does not rely on ground-truth labels, it can be pre-trained on a combination of several large-scale person datasets after extracting their keypoints. We speculate that employing multi-dataset co-training can further improve the performance of HAP, as observed in [78]. We also recognize that our HAP would derive great benefits from more accurate keypoints (*e.g.*, human-annotated keypoints) if available. We will discuss this point in details later.

Table 5: **Hyper-parameters of pre-training.** HAP follows most of the settings of MAE and CAE.

| dataset | method | batch size | epochs | warmup epochs | base learning rate | optimizer | weight decay |
|---------|--------|-----------|--------|---------------|-------------------|-----------|--------------|
| LUPerson | MAE [30] | 4096 | 400 | 40 | 1.5e-4 | AdamW | 0.05 |
| LUPerson | CAE [30] | 2048 | 400 | 10 | 1.5e-4 | AdamW | 0.05 |

**Implementations details.** We follow most of the training settings of MAE/CAE except using training epochs of 400 (Table 5). Different from MAE/CAE, HAP has a large scaling ratio of 80%, and an intermediate masking ratio of 50% with block-wise masking strategy, as introduced in the main paper.

# B   Details of human-centric perception tasks

We list the training details of human-centric perception tasks in Table 1 of the main paper as follows.

## B.1   Person ReID

Table 6: **Hyper-parameters of person ReID.**

| dataset | batch size | epochs | learning rate | optimizer | weight decay | layer decay | drop path |
|---------|-----------|--------|---------------|-----------|--------------|-------------|-----------|
| MSMT17 | 64 | 100 | 8e-3 | AdamW | 0.05 | 0.4 | 0.1 |
| Market-1501 | 64 | 100 | 8e-3 | AdamW | 0.05 | 0.4 | 0.1 |

We consider two ReID datasets: MSMT17 [74] (Figure 8b) and Market-1501 [87] (Figure 8c) with the codebase [4] of [50]. The input image resolutions are set to $256 \times 128$ and $384 \times 128$. Data augmentations are the same as [50], including resizing, random flipping, padding, and random cropping. Following [32, 64, 8, 16], the model is fine-tuned using a cross-entropy loss and a triplet loss with equal weights of 0.5. We also consider using a convolution-based module [50] to further improve results. The evaluation metric of mAP (%) is reported. See hyper-parameters in Table 6.

## B.2   2D human pose estimation.

HAP is evaluated on MPII [1] (Figure 8d), MS COCO [46] (Figure 8e), and AI Challenger (AIC) [75] (Figure 8f). We use the codebase [5] of [78]. Resolutions of $256 \times 192$ and $384 \times 288$ are employed.

---

[3]In this paper, we conduct all experiments on base-sized model to verify the effectiveness of our method. In fact, all scales of models are compatible with HAP.

[4]https://github.com/YanzuoLu/MALE

[5]https://github.com/ViTAE-Transformer/ViTPose

Table 7: **Hyper-parameters of 2D pose estimation.** $^{\dagger}/^{+}$: multi-dataset training/$384 \times 288$ resolution.

| dataset | batch size | epochs | learning rate | optimizer | weight decay | layer decay | drop path |
|---|---|---|---|---|---|---|---|
| MPII/MPII$^{+}$ | 512 | 210 | 1.0e-4 | Adam | - | - | 0.30 |
| COCO/COCO$^{+}$ | 512 | 210 | 2.0e-4 | AdamW | 0.1 | 0.80 | 0.29 |
| AIC/AIC$^{+}$ | 512 | 210 | 5.0e-4 | Adam | - | - | 0.30 |
| MPII$^{\dagger}$/COCO$^{\dagger}$/AIC$^{\dagger}$ | 1024 | 210 | 1.1e-3 | AdamW | 0.1 | 0.77 | 0.30 |
| MPII$^{\dagger+}$/COCO$^{\dagger+}$/AIC$^{\dagger+}$ | 512 | 210 | 5.0e-4 | AdamW | 0.1 | 0.75 | 0.30 |

Data augmentations are random horizontal flipping, half body transformation, and random scaling and rotation. We follow the common top-down setting, i.e., estimating keypoints of the instances detected [76]. We use mean square error (MSE) [78] as the loss function to minimize the difference between predicted and ground-truth heatmaps. Following [78], we also perform multi-dataset training setting by co-training on MPII, MSCOCO, and AIC. Evaluation metric of PCKH (%) is reported for MPII, and AP (%) is reported for MS COCO and AIC. See the details of hyper-parameters in Table 7.

## B.3 Text-to-image person ReID

Table 8: **Hyper-parameters of text-to-image person ReID.**

| dataset | batch size | epochs | learning rate | optimizer |
|---|---|---|---|---|
| CUHK-PEDES | 64 | 60 | 1e-3 | Adam |
| ICFG-PEDES | 64 | 60 | 1e-3 | Adam |
| RSTPReid | 64 | 60 | 1e-3 | Adam |

We employ three commonly used datasets, *i.e.*, CUHK-PEDES [42] (Figure 8g), ICFG-PEDES [20] (Figure 8h), RSTPReid [89] (Figure 8i). The codebase [6] of [59] is used. The resolution is $384 \times 128$. We adopt resizing and random horizontal flipping as data augmentations. BERT [19] is used to extract embeddings of texts, which are then fed to Bi-LSTM [27]. Feature dimensions of images and texts are both set to 384. The BERT is frozen while Bi-LSTM and ViT are fine-tuned. We report Rank-1 (%) metric for the results on all datasets. The details of the hyper-parameters can be found in Table 8.

## B.4 3D pose and shape estimation

Table 9: **Hyper-parameters of 3D pose and shape estimation.**

| dataset | batch size | epochs | learning rate | optimizer |
|---|---|---|---|---|
| 3DPW | 192 | 11 | 1e-4 | SGD |

We perform 3D pose and shape estimation on 3DPW [67] (Figure 8j) using the codebase [7] of [15]. The resolution is $256 \times 256$. Data augmentations follow [15]. We minimize a pose loss, *i.e.*, $l_1$ distance between the predicted 3D joint coordinates and the pseudo ground-truth, and a mesh loss, *i.e.*, loss for predicted SMPL parameters [53, 55]. We report MPJPE, PA-MPJPE, MPVPE. The former two are for 3D pose, while the latter one is for 3D shape. Hyper-parameters are in Table 9.

## B.5 Pedestrian attribute recognition

We evaluate HAP on three popular datasets, *i.e.*, PA-100K [47] (Figure 8k), RAP [39] (Figure 8l), and PETA [18] (Figure 8m). We implement it using the codebase [8] of [36]. The resolution is $256 \times 192$. Data augmentations consist of resizing and random horizontal flipping. The binary cross-entropy loss is employed, and the evaluation metric of mA (%) is reported. Find the hyper-parameters in Table 10.

---

[6] https://github.com/Galaxfy/ReID-test1
[7] https://github.com/hongsukchoi/3DCrowdNet_RELEASE
[8] https://github.com/valencebond/Rethinking_of_PAR

Table 10: **Hyper-parameters of pedestrian attribute recognition.**

| dataset | batch size | epochs | learning rate | schedule | optimizer | weight decay |
|---|---|---|---|---|---|---|
| PA-100K | 64 | 55 | 1.7e-4 | cosine | AdamW | 5e-4 |
| RAP | 64 | 8 | 2.4e-4 | cosine | AdamW | 5e-4 |
| PETA | 64 | 50 | 1.9e-4 | cosine | AdamW | 5e-4 |

## C   Study of human part prior

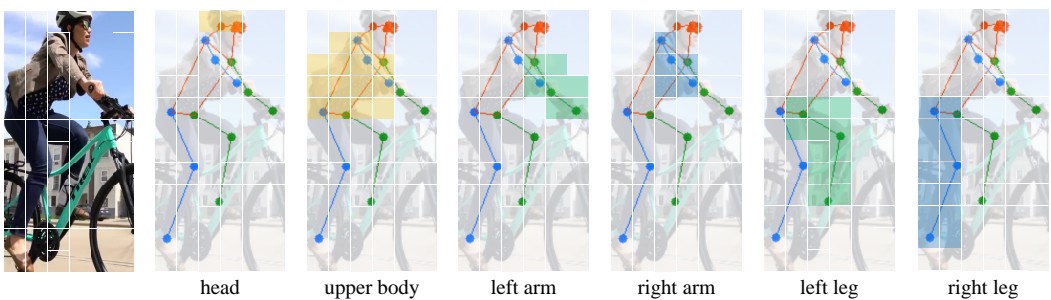

head     upper body     left arm     right arm     left leg     right leg

Figure 7: **Part masking with keypoints.** We divide human body structure into 6 parts following [60]. For each input image, we randomly select 0 to 6 parts (following a uniform distribution) and mask patches located on them (according to the sequence of human part selection). For each selected part, we mask the patches within the bounding boxes of the corresponding keypoint pairs, *i.e.*, *head*: {(nose, left eye), (nose, right eye), (left eye, right eye), (left eye, left ear), (right eye, right ear)}; *upper body*: {(left shoulder, right hip), (right shoulder, left hip)}; *left arm*: {(left shoulder, left elbow), (left elbow, left wrist)}; *right arm*: {(right shoulder, right elbow), (right elbow, right wrist)}; *left leg*: {(left hip, left knee), (left knee, left ankle)}; *right leg*: {(right hip, right knee), (right knee, right ankle)}.

**Details of part masking with keypoints.** We incorporate human parts, a human structure prior, into pre-training. Figure 7 illustrates the details of how to mask each of the six body parts with keypoints.

Table 11: **Guidance** in mask sampling. Baseline uses no guidance, *i.e.*, random block-wise masking.

| guidance | MSMT17 | MPII |
|---|---|---|
| attention map | 71.7 | 90.4 |
| keypoint | 75.9 | 91.6 |
| part | **76.4** | **91.8** |
| baseline | 75.6 | 91.6 |

**Masking guidance in mask sampling strategy.** We study different guidance, reported in Table 11. Some MIM works [83, 29, 6, 37] use attention map to identify and mask the informative image patches. However, this design performs poorly in human-centric pre-training, achieving even lower performance than baseline (71.7 *vs.* 75.6 on MSMT17). We conjecture that the underlying reason is the challenge for the model to obtain precise and meaningful attention maps, especially in the initial stage of the pre-training process. That may be why recent works [40, 37, 6] start to employ strong pre-trained models to assist mask sampling. In contrast, we directly employ intuitive and explicit human part prior, *i.e.*, human keypoints, to guide mask sampling, which shows great potential. Besides, we observe that only masking patches located on keypoints, instead of human parts, improves little. It indicates that masking a whole human body works better, similar to the finding that block-wise masking performs better than random masking in human-centric pre-training (see the main paper).

**Part selection.** Here, we ablate different part selection strategies in Table 12. We observe that selecting a fixed number of parts is not beneficial. Compared to a whole range, selecting within a small range has a little improvement. It shows that more part variations brings more randomness, which is good for pre-training. Similar to the findings in recent studies [83, 29, 6, 37] that a deliberate

Table 12: **Part selection** for masking. Baseline uses no guidance, *i.e.*, random block-wise masking.

| part selection | MSMT17 | MPII |
|---|---|---|
| 3 | 75.4 | 91.5 |
| 6 | 74.4 | 91.0 |
| unif$\{0,3\}$ | 76.1 | 91.7 |
| unif$\{0,6\}$ | **76.4** | **91.8** |
| baseline | 75.6 | 91.6 |

design of mask sampling strategy would have a positive impact, our part selection strategy is beneficial for human-centric pre-training, resulting in great performance on a range of downstream tasks.

**Using keypoints as target.** Since keypoints can also be used as supervision, we perform this baseline by making the decoder predict keypoints instead of pixels. The obtained results are 50.3% on MSMT17 and 89.8% on MPII with 100-epoch pre-training, while our HAP is 72.2% and 91.3%. The possible reason is that regressing keypoints may lose informative details provided by pixels, which has negative impact on the tasks. In contrast, HAP incorporates prior knowledge of structure into pre-training, which is beneficial to maintain both semantic details and information of body structure.

**Using another keypoint detector.** To study the robustness of HAP to the extracted keypoints, we make a simple exploration by employing another off-the-shelf keypoint detector named OpenPose [3]. It achieves 72.0% on MSMT17 and 91.2% on MPII with 100-epoch pre-training. The results are comparable to our HAP with ViTPose [78] as the pose detector (72.2% and 91.3% on MSMT17 and MPII respectively), demonstrating that our HAP is generally robust to the keypoint detector method.

## D  Study of structure-invariant alignment

Table 13: Analysis of **loss weight** $\gamma$. $\gamma = 0$ represents our baseline that does not use alignment loss.

| $\gamma$ | MSMT17 | MPII |
|---|---|---|
| 0.5 | 75.6 | 91.2 |
| 0.1 | 76.3 | **91.8** |
| 0.05 | **76.4** | **91.8** |
| 0.01 | 76.0 | 91.6 |
| 0 (baseline) | 75.1 | 91.1 |

**Loss weight** $\gamma$**.** Table 13 demonstrates that the weight $\gamma$ of Eq.2 is beneficial across a wide range.

Table 14: **Type of two views in alignment.** The "feature 1" and "feature 2" are aligned via $\mathcal{L}_{\mathrm{align}}$.

| feature 1 | feature 2 | MSMT17 | MPII |
|---|---|---|---|
| masked view | global view | 74.5 | 91.0 |
| masked view | visible view | 76.1 | 91.7 |
| masked view | masked view | **76.4** | **91.8** |
| baseline | | 75.1 | 91.1 |

**Type of two views in alignment.** By default, we use two random masked views for alignment. We further explore other two types of alignment, see Table 14. Aligning the masked view with the global view (*i.e.*, without masking) shows poor performance. We attribute it to the high degree of overlap between the two views, which makes the task too easy. Aligning the masked view and the visible view with no overlap yields improvement, but still inferior than the default. It shows that learning subtle humans characteristics by aligning random views of corrupted body structure is important.

**Removing negative samples in the alignment loss.** We use the loss of SimSiam [11] instead. The results are 76.1% (SimSiam) vs. 76.4% (ours) on MSMT17, 91.6% (SimSiam) vs. 91.8% (ours) on MPII. It demonstrates that our HAP still works well if removing the negatives in the alignment loss.

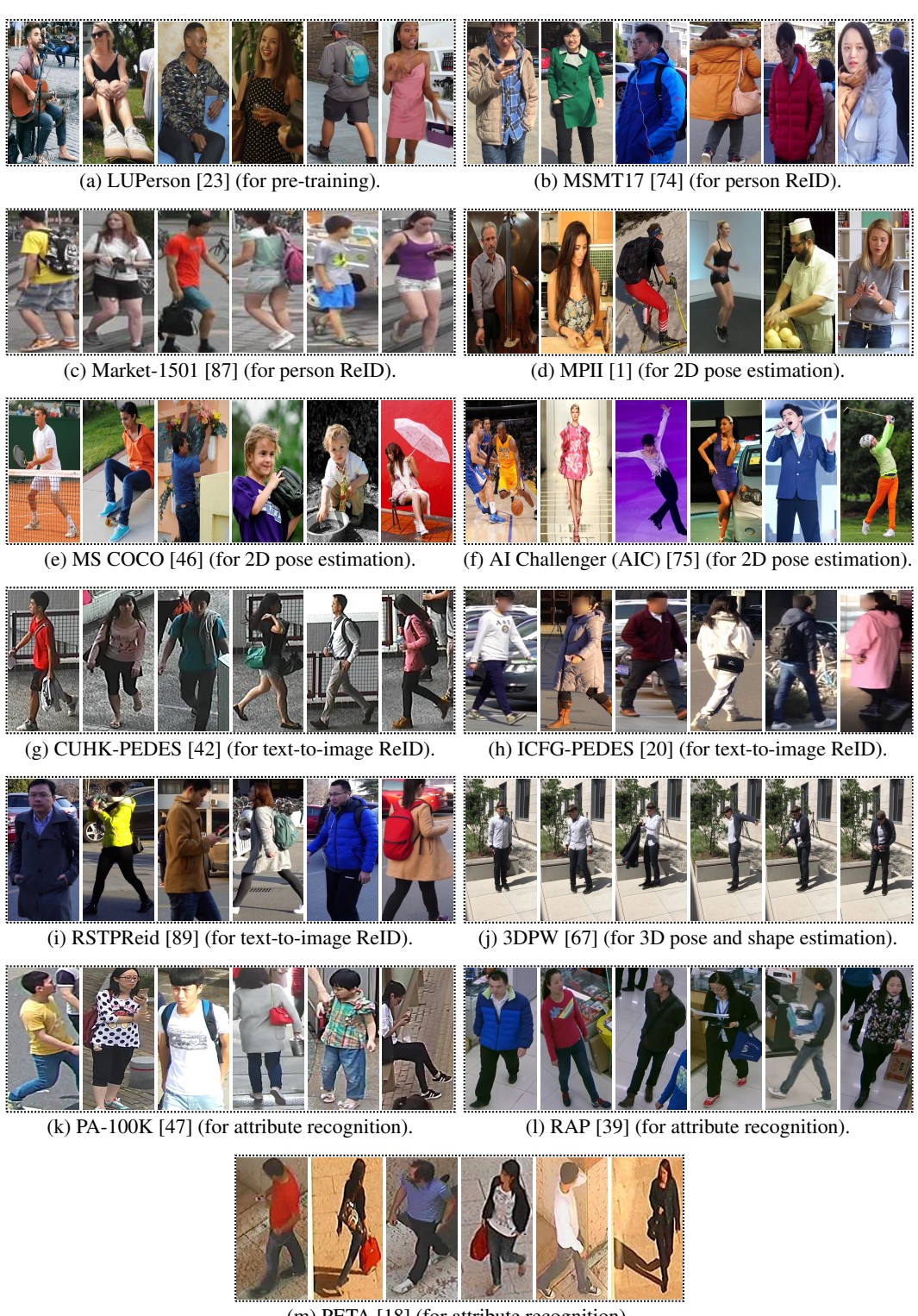

Figure 8: **Example images of the used datasets.** Our proposed HAP is pre-trained on (a) LUPerson, and evaluated on a broad range of tasks and datasets, including (b-c) person ReID, (d-f) 2D pose estimation, (g-i) text-to-image person ReID, (j) 3D pose and shape estimation, (k-m) pedestrian attribute recognition. These datasets encompass diverse environments with varying image quality, where persons are varied by scales, occlusions, orientations, truncation, clothing, appearances, and postures, demonstrating the the great generalization ability of HAP in human-centric perception.

