# OpenReview forum: "HAP: Structure-Aware Masked Image Modeling for Human-Centric Perception"
_NeurIPS.cc/2023/Conference — NeurIPS 2023 poster_

### Official Review · Reviewer_rHqg · 2023-07-01

**Soundness:** 3 good
**Presentation:** 3 good
**Contribution:** 3 good
**Rating:** 6
**Confidence:** 5

**Summary:**

This paper proposes a Human structure-Aware Pre-training (HAP) method that incorporates human structure priors into the masked image modeling (MIM) training strategy [26] for tasks related to human-centric perception. The authors have demonstrated the advantages of the proposed method on 5 human-centric perception tasks across 12 benchmark datasets. Overall, the method is simple and intuitive, and extensive experiments have provided solid evidence of its superiority in human-centric perception tasks.

**Strengths:**

1. The author, through analyzing the deficiencies of the current MIM training strategy [26] in human-centric perception tasks, further proposes the introduction of human structure priors to expand the MIM training approach. This method is intuitive and appropriate.

2. Although the author uses the existing InfoNCE loss to implement structure-invariant alignment loss, the application of this structure-invariant alignment concept to different masked views in order to align them, thereby enhancing the feature representation ability of human structure information, is both straightforward and promising.

3. The author not only demonstrated the superiority of the proposed method across twelve benchmark datasets, but also validated the benefits of each proposed training strategy or loss function through ablation experiments.

4. The organization of the article and the presentation of the methods are both clear and well-structured.


**Weaknesses:**

While this method effectively integrates prior knowledge and existing loss functions to enhance the performance of human-centric perception tasks, its technical originality and novelty still leave something to be desired.



**Questions:**

1. The accuracy of pose estimation methods seems to significantly impact the performance of the HAP proposed by the author. However, it appears that the author hasn't conducted sufficient analysis, such as how OpenPose or AlphaPose specifically affect the performance of HAP.

2. In 3D human pose and shape estimation task, the performance of 3D human mesh estimation often decreases due to occlusions. The HAP method proposed by the author seems promising in solving this issue. However, it's unfortunate that the author did not conduct relevant experiments to further highlight the advantages of the proposed HAP. It would be beneficial for the author to incorporate such experiments to enhance the strengths of their HAP.






**Limitations:**

The author mentions that their proposed method may have potential negative societal impacts when applied to image generation tasks. Personally, I consider this to be a problem that we collectively need to address in the process of advancing artificial intelligence. Therefore, it does not diminish my positive evaluation of the research work. However, I still hope that the author can provide possible solutions to mitigate these potential negative impacts.

---

> ### Author Rebuttal · Authors · 2023-08-10
>
> **Q1: The accuracy of pose estimation methods seems to significantly impact the performance of the HAP proposed by the author. However, it appears that the author hasn't conducted sufficient analysis, such as how OpenPose or AlphaPose specifically affect the performance of HAP.**
>
> R1: Thanks for pointing out this. We further experiment HAP with 2D keypoints extracted by OpenPose.
> The results are 72.0\%/91.2\% on MSMT17/MPII with 100-epoch per-training, which are similar to the results with the keypoints extracted by ViTPose (our original HAP), i.e., 72.2\%/91.3\% on MSMT17/MPII.
> They both achieve superior performance than baseline (69.4\% on MSMT17 and 90.4\% MPII).
> It reflects that the accuracy of pose estimation has slight effect on the pre-training quality.
> We will further study more pose estimation methods to validate our statement.
>
> **Q2: In 3D human pose and shape estimation task, the performance of 3D human mesh estimation often decreases due to occlusions. The HAP method proposed by the author seems promising in solving this issue. However, it's unfortunate that the author did not conduct relevant experiments to further highlight the advantages of the proposed HAP. It would be beneficial for the author to incorporate such experiments to enhance the strengths of their HAP.**
>
> R2: Thanks for your constructive suggestion.
> We will add occlusion-based experiments in 3D human pose and shape estimation task by carefully studying experimental settings in the revised version to further enhance the strengths of our HAP.
>
> **Q3: The author mentions that their proposed method may have potential negative societal impacts when applied to image generation tasks. Personally, I consider this to be a problem that we collectively need to address in the process of advancing artificial intelligence. Therefore, it does not diminish my positive evaluation of the research work. However, I still hope that the author can provide possible solutions to mitigate these potential negative impacts.**
>
> R3: Thanks a lot for your positive evaluation of our research work.
> And very thanks for pointing out the suggestion that we can "provide possible solutions to mitigate these potential negative impacts''.
> We will carefully update the "Broader impacts'' section as follows based on your suggestion in the revised version.
>
> "Broader impacts and possible solutions: The proposed method is pre-trained and evaluated on person datasets, thus could reflect biases in these datasets, including those with negative impacts. When using the proposed method and the generated images, one should pay careful attention to dataset selection and pre-processing to ensure a diverse and representative sample. To avoid unfair or discriminatory outcomes, please analyze the model's behavior with respect to different demographics and consider fairness-aware loss functions or post-hoc bias mitigation techniques. We advocate for responsible use, and encourage the development of tools to detect and mitigate the potential misuse of our HAP and other masked image modeling techniques for unethical purposes."

---

> > ### Comment · Reviewer_rHqg · 2023-08-15
> >
> > The author has responded thoroughly to my questions. However, it's regrettable that I have yet to see results from experiments based on occlusion. Nonetheless, I still hold a positive view of this paper and look forward to the author supplementing with occlusion experiments and making the model public to drive community development. I will vote in favor of accepting this submission.

---

> > > ### Author Response · Authors · 2023-08-18
> > >
> > > We greatly appreciate your thoughtful assessment on our response and your positive view of our work.
> > >
> > > We also apologize for not including the occlusion experiments up to now.
> > > We are actively working on the occlusion experiments, unfortunately, we need more time to comprehensively understand the experimental setting to ensure that the experiments are correct and the comparison is fair.
> > > We agree with you that the occlusion experiments can further enhance the strengths of our HAP, and we will keep going on this exploration.
> > >
> > > We will make the model and code public following your suggestion and expectation, and we also hope that our publicly available model and code can drive the community development.
> > >
> > > Thanks again for your positive comments which are valuable to improve the quality and impact of our work!

---

### Official Review · Reviewer_ueQY · 2023-07-04

**Soundness:** 3 good
**Presentation:** 3 good
**Contribution:** 2 fair
**Rating:** 5
**Confidence:** 5

**Summary:**

The authors introduce masked image modeling as a pre-training method specifically designed for human-centric perception tasks. To this end, the authors incorporate human structure priors (high scaling ratio, mediate masking ratio, block-wise masking), human part prior (2D keypoints as guidance for mask sampling), and structure-invariant alignment loss (contrastive loss on the [CLS] tokens across views). Extensive experiments demonstrate that the proposed method achieves competitive performance on various human-centric perception tasks.

**Strengths:**

-	The proposed method is well motivated. Directly applying MIM in human-centric perception tasks will indeed induce some problems. The authors exploited some priors from the person dataset to make pre-training tailored for specific downstream tasks.
-	The paper is generally well-written and easy to follow.
-	The experiments are extensive and the results are promising.


**Weaknesses:**

-	The technical novelty is limited. The proposed method is essentially a combination of many previous techniques used in masked image modeling. For example, the block-wise masking has been proposed in BEiT [2]. Semantic-guided masking has been proposed in prior works [5, 25, 33, 36, 41, 60, 74]. The only difference is that these works use attention maps while the authors use keypoints. Adding alignment loss is actually combining masked image modeling with contrastive learning (like what has been done in iBOT [77]).
-	The authors use 2D keypoints to guide pre-training, which are extracted by existing pose estimation methods. I am afraid the obtained keypoints are not fully unsupervised.


**Questions:**

I have an additional question. The authors use contrastive loss with negative samples. What about removing negatives? Prior works (e.g., BYOL, SimSiam, DINO) have proven that negative samples are not necessary for better representation learning. Overall, I am leaning towards borderline accept considering that the authors provide a new formulation of previous techniques for a new human-centric perception scenario.

**Limitations:**

The authors have discussed the limitations and the broader impacts in Sec. 5, which looks good to me.

---

> ### Author Rebuttal · Authors · 2023-08-10
>
> **Q1: The technical novelty is limited. The proposed method is essentially a combination of many previous techniques used in masked image modeling. For example, the block-wise masking has been proposed in BEiT [2]. Semantic-guided masking has been proposed in prior works [5, 25, 33, 36, 41, 60, 74]. The only difference is that these works use attention maps while the authors use keypoints. Adding alignment loss is actually combining masked image modeling with contrastive learning (like what has been done in iBOT [77]).**
>
> A1: Thanks, we agree that many techniques have been used in masked image modeling (MIM), while
> our HAP further **revisits and reformulates** these techniques in a novel way to introduce **human structure priors** (our key contribution) into MIM pre-training for human-centric perception tasks. In details,
>
> i) Block-wise masking.
> Using block-wise masking (proposed in BEiT [2]) is just a **finding** that it performs sightly better than random-wise masking in human-centric perception tasks (refer to preliminary study in Sec. 3.1), thus it is not the main point in our HAP.
>
> ii) Human part prior guided masking is different from semantic-guided masking.
> The proposed human part prior guided masking is the key point in HAP that is **specialized** for human-centric tasks by introducing keypoints.
> It can be regarded as an **external and human-friendly** guidance for masking, instead of **self** guidance from attention maps in semantic-guided masking [5, 25, 33, 36, 41, 60, 74].
> We also experiment HAP by directly using attention maps to guide masking.
> This setting achieves 71.7\%/90.4\% on MSMT17/MPII (refer to Table 7 in Appendix), largely inferior than our HAP with human part guidance (76.4\%/91.8\%) and with keypoints (75.9\%/91.6\%),
> which evidently shows the superiority of the proposed masking strategy guided by human prior.
>
> iii) Adding alignment loss combines MIM with contrastive learning.
> We agree. Despite similar formulation as contrastive loss, the proposed structure-aware alignment concept further applies contrastive loss on masked views generated from human part prior to enhance the feature discriminative ability.
> iBOT [77] is different from ours in: a) the masked views are randomly generated without prior guidance. b) the loss is formulated by cross-entropy loss, not contrastive loss as in ours.
>
> **Q2: The authors use 2D keypoints to guide pre-training, which are extracted by existing pose estimation methods. I am afraid the obtained keypoints are not fully unsupervised.**
>
> R2: Thanks for pointing out this concern.
> The 2D keypoints only provides the prior guidance for mask sampling in HAP, rather than the supervision signal for pre-training.
> HAP only use image pixels as the pre-training targets, and it is a self-supervised learning method.
> We will carefully revise the manuscript to make it clear.
>
> **Q3: The authors use contrastive loss with negative samples. What about removing negatives? Prior works (e.g., BYOL, SimSiam, DINO) have proven that negative samples are not necessary for better representation learning.**
>
> Thanks for your constructive suggestion.
> Following your advice, we perform our HAP using the loss of SimSiam to replace our original alignment loss.
> The results are 76.1\% (loss of SimSiam) vs. 76.4\% (our alignment loss) on MSMT17 for Person ReID, 91.6\% (loss of SimSiam) vs. 91.8\% (our alignment loss) on MPII for 2D pose estimation.
> It shows that our HAP still works well if removing the negatives in contrastive loss, while our alignment loss with negatives performs slightly better.
> We will include this interesting finding in the revised version.

---

> > ### Comment · Reviewer_ueQY · 2023-08-16
> > **Thanks for the response**
> >
> > Thanks for the authors' rebuttal. My concerns are largely addressed. Although the technical novelty is somewhat limited, I am in favor of accepting this submission considering the proposed method provides a new formulation for the human-centric perception scenario, which may have a positive impact on the human-centric vision community. For Q2, I understand that HAP only uses image pixels as the pre-training target. However, the introduction of 2D keypoints still needs supervision unless the corresponding pose estimation methods are unsupervised. The authors should be careful when claiming their method as "self-supervised".

---

> > > ### Author Response · Authors · 2023-08-17
> > >
> > > Thanks for your response and suggestion.
> > > We greatly appreciate your positive feedback regarding our work and your approval about the positive impact of our work on the human-centric vision community.
> > > We understand and agree your concern that "the introduction of 2D keypoints still needs supervision unless the corresponding pose estimation methods are unsupervised".
> > > We thus will carefully revise our manuscript about the "self-supervised" claim following your suggestion.
> > > Thanks again!

---

### Official Review · Reviewer_BM1x · 2023-07-06

**Soundness:** 3 good
**Presentation:** 4 excellent
**Contribution:** 3 good
**Rating:** 5
**Confidence:** 4

**Summary:**

The work studies masked image modeling (MIM) in human-centric perception. It first revisits the vanilla MIM and finds that human structure prior (2D pose) helps the downstream human-related tasks. This encourages the authors to incorporate this prior into the classical MAE. Based on this human-centric masking strategy, a structure-invariant alignment loss is developed as a regularization for pre-training. To evaluate the effectiveness of the proposed method, the authors conduct extensive experiments on 11 human-centric benchmarks. The competitive performance is observed when compared to the recent human-centric pre-training counterparts.

**Strengths:**

1) The overall writing and organization are satisfactory. The paper well describes the necessity of human structure prior to human-centric pre-training. The statistics are convincing to convey the motivation to the audience.

2) The pipeline incorporates MAE with pose-related masking is simple and effective. It can be easily applied to any MIM method.

3) The experiments are sufficient to verify the effectiveness of the proposed method, with informative visualization and ablation studies.

4) Codes are attached, and implementation details are well described for re-implementation.

**Weaknesses:**

The performance on multiple benchmarks is overclaimed. For example, the results of Attribute recognition are marginal when comparing HAP with other human-centric pre-training methods. With bells and whistles, HAP (multi-dataset training and larger image size) outperforms other methods significantly. However, the vanilla HAP is inferior to the LiftedCL. PATH, and UniHCP, though they use different pre-training datasets. So, I reckon the superior performance this paper claims should be re-examined, or restated at least.



**Questions:**

As the technical novelty of this paper is human-related masking, the authors are encouraged to ablate the part selection. For example, the P parts are randomly selected, as denoted in Section 3.2. I am wondering how the performance will be if only making one or two parts, such as the head or upper body. Then that will be six ablation studies (each one masks a part only) to investigate the impact of each part.

**Limitations:**

In the limitation section, the authors state that the accuracy of pose estimation could affect the pre-training quality. However, the quantitative or qualitative results are not presented.

---

> ### Author Rebuttal · Authors · 2023-08-10
>
> **Q1: The performance on multiple benchmarks is overclaimed. For example, the results of Attribute recognition are marginal when comparing HAP with other human-centric pre-training methods. With bells and whistles, HAP (multi-dataset training and larger image size) outperforms other methods significantly. However, the vanilla HAP is inferior to the LiftedCL, PATH, and UniHCP, though they use different pre-training datasets. So, I reckon the superior performance of this paper claims should be re-examined, or restated at least.**
>
> R1: Thanks for your valuable advice. We will re-examine and restate the claim of superior performance in the revised version following your advice.
> For clarification, our HAP is a self-supervised method, while PATH and UniHCP are supervised learning method, thus the comparison is not totally fair.
> When comparing with other self-supervised methods (like SOLIDER), our HAP achieves superior performance.
>
> **Q2: I am wondering how the performance will be if only making one or two parts, such as the head or upper body. Then that will be six ablation studies (each one masks a part only) to investigate the impact of each part.**
>
> Thanks for this intriguing point. Following your suggestion, we implement six ablation studies, in which each one only masks one part, i.e., head, left arm, right arm, left leg, right leg and upper body, to investigate the impact of each part.
> Other settings keep the same as in overall HAP.
> The results in the below table show that our overall HAP achieves slightly better performance than only masking one part.
> Among six parts, the upper body part can provide relatively more information than others.
>
> | dataset | unif{0, 6} | head | left arm | right arm | left leg | right leg | upper body |
> | --- | --- | --- | --- | --- | --- | --- | --- |
> | MSMT17 | 72.2 | 71.8 | 72.0 | 71.9 | 71.9 | 72.0 | 72.1 |
> | MPII | 91.3 | 90.9 | 91.2 | 91.1 | 91.1 | 91.1 | 91.2 |
>
> **Q3: In the limitation section, the authors state that the accuracy of pose estimation could affect the pre-training quality. However, the quantitative or qualitative results are not presented.**
>
> R3: Thank you for pointing out this.
> We add an experiment that uses 2D keypoints extracted from OpenPose, another pose estimation method, for the pre-training of HAP.
> It achieves 72.0\%/91.2\% on MSMT17/MPII with 100-epoch pre-training, which are similar to our original HAP with ViTPose as pose detector (72.2\%/91.3\% on MSMT17/MPII).
> It reflects that the accuracy of pose estimation has slight effect on the pre-training quality.
> We will further study more pose estimation methods to validate our statement.

---

> > ### Comment · Reviewer_BM1x · 2023-08-14
> >
> > My concerns have been well addressed. For example, the ablation of different body parts is provided with a convincing analysis. I will vote for acceptance for this submission.

---

> > > ### Author Response · Authors · 2023-08-14
> > > **Response to Reviewer BM1x (2)**
> > >
> > > We really appreciate your positive feekback on our work and response.
> > > We will carefully revise our paper following your suggestions and comments, including adding the ablation and the analysis of different body parts. Thanks!

---

### Official Review · Reviewer_QQMw · 2023-07-06

**Soundness:** 3 good
**Presentation:** 3 good
**Contribution:** 2 fair
**Rating:** 6
**Confidence:** 5

**Summary:**

This paper propose a pertaining strategy for human-centric vision tasks. They extend the mask image modelling approach by incorporating a prior on the human body parts to guide the mask sampling strategy. In short, they mask parts of the image which contains body part. Authors also propose an alignement loss to make sure that the same image with two different masks have same feature representations. They show fine-tuning performance on several standard benchmarks.

**Strengths:**

1) The paper is well-written and easy to read. The figures are clear and they help to get the main idea of the paper. Since authors do not claim to bring a very novel idea/technical contribution they explain that they start from an existing work (MAE) and try to extend this method for human-centric tasks. They perform experiments to identify issues for this method and they propose patches to fix them. The intuition behind the proposed fixs are based on human prior and are explained with references to prior works (i.e. body parts). Even if we could argue that the novelty is somehow limited, the proposed paper is well-constructed and may have a positive impact in the vision community specially for researchers working on human0centric vision.
2) The introduction introduced well the research problem, and summarise the main idea and claim of the paper. I appreciate that authors do not claim their method as SSL (even if they do in the Appendix l.13) because they use pseudo-GT (keypoints) for sampling masks in the input images.
3) Section 3.1 is very much appreciated. I guess that the project started with these experiments so it make sens t.
4) Authors presented good ablation studies (Table 2-3), as a reader we understand the impact of each components during the pretraing (some important ones are still missing see weaknesses).
5) Supplementary materials contain all the details for each downstream tasks which make the results reproducible. Furthermore authors also include the code for pretraining using their approach. Authors mentioned that code will be release soon.
6) Good performance on several downstream benchmarks where they compare against recent works. They reach SoTA on multiple datasets.

**Weaknesses:**

1) One major weakness of this submission is the lack of a simple baseline which is; pretraining on LUPerson by training to regress the 2d keypoints extracted by VITPose. I understand that they are noisy because they are pseudo-GT but I guess that because there are more than 2M images in LUPerson it should already give a better understand of why we need to pretrain using supervision signal from the pixel instead of using supervision signal for the 2d keypoints. For example for the downstream tasks of 3D human mesh recovery most of the training data used by the research community are pseudo-GT extracted on in the wild images and it works better than using indoor image with perfect ground-truth.
2) Authors mentioned that 2d keypoints are obtained by VITPose, it would be great to study the impact of the keypoint quality by using another off-the-shell detector such as OpenPose or pretraining on a big dataset with ground-truth 2D pose such as MSCOCO. I think that such ablation is very important for making sure that the key component of your proposed method is: a) to leverage a large-scale dataset for pretrainning — or b) to use a very recent off-the-shell 2d pose detector. At the moment we do not have any answer to this question in the proposed manuscript. I would suggest to pretrain your method on either LUPerson or MSCOCO with the same number of pretraining examples and using either GT 2d pose of 2D pose extracted from VITPose.
3) I did not find the information in the paper of Supp. Mat. About the initialisation of the weights for the pretraining stage. Do you train from scratch or from MAE-Imagenet?
4) It is a bit surprising that the fine-tuning on the downstream task of 3D pose does not work as well as [14] since you are just changing the backbone. It is a bit counter-intuitive with the results that you get on the other tasks. Moreover the backbone in [14] is a CNN pertained on 2d pose estimation so the conclusion regarding this task are a bit weird in comparison to the other tasks.
5) The Structure Alignement Loss is appealing since it bring contrastive learning in the paper but given results in Table 2 it seems that it does not bring a significant gain. The improvements are quite marginal and similar to HAP-2 given a certain variance during the fine-tuning stage (only on MSMT17 results are better than HAP-2).
6) Other human-centric pretraining method such as PATH are proposing zero-shot or results on downstream task with a frozen encoder. Do you have some results following this fine-tuning strategy (i.e. training only the head). It would be great to see how important it is to fine-tune the encoder for each task. Or maybe using adaptor.

**Questions:**

1) l.44 “Or a given input image, we generate two views. Not sure that we can really say that there are two views. Using the word ‘view’ makes reference to 3D and here the masked images are from the same image not from two different camera viewpoints. So I would suggest to find a different explaination.
2) Most of my questions are in the weaknesses. Weaknesses 1) and 2) are the most important points to me.

---

> ### Author Rebuttal · Authors · 2023-08-10
>
> **Q1: Lack of a simple baseline: pre-training on LUPerson by training to regress the 2D keypoints extracted by ViTPose.**
>
> A1: Thanks for your constructive suggestion. We experiment this simple baseline and achieve 50.3\% on MSMT17 and 89.8\% on MPII with 100-epoch pre-training. This baseline is significantly inferior than our original HAP with RGB pixel as supervision signal (72.2\% on MSMT17 and 91.3\% on MPII). The underlying reason is that regressing 2D keypoints may lose informative details provided by RGB pixels, which has negative impact on human-centric tasks, especially on ReID tasks (refer to MSMT17 results). In contrast, our HAP is able to incorporate prior knowledge of human body structure into the pre-training, which is beneficial to maintain both details and body-structure information. We will add the above experiment and analysis in the revised version.
>
> **Q2: It would be great to study impact of the keypoint quality by using another off-the-shell detector such as OpenPose or pre-training on a big dataset with ground-truth 2D pose such as MSCOCO.**
>
> A2: Thanks for your advice.
>
> i) Using 2D keypoints extracted by another off-the-shell detector OpenPose in HAP achieves 72.0\% on MSMT17 and 91.2\% on MPII with 100-epoch pre-training. The results are similar to our original HAP with ViTPose as pose detector (72.2\%/91.3\% on MSMT17/MPII), reflecting that our method is robust to the off-the-shell detector method.
>
>
> ii) MSCOCO has about **260k** images for the 2D pose task. Using MSCOCO with its 2D ground-truth pose in HAP achieves 60.3\%/89.5\% on MSMT17/MPII with 100-epoch pre-training. Moreover, using the same number of pre-training samples of LUPerson (about 260k) achieves 63.0\%/89.7\%. These results are inferior than our original HAP with **2.1M** LUPerson (72.2\%/91.3\% on MSMT17/MPII), showing that large-scale dataset is important for pre-training.
>
> **Q3: I did not find the information in the paper of Supp. Mat, about the initialization of the weights for the pre-training stage. Do you train from scratch or from MAE-ImageNet?**
>
> A3: Our HAP is trained with the weight initialized from MAE-ImageNet (Line 18-19 in Appendix).
>
> **Q4: The conclusion regarding 3D pose task area bit weird in comparison to the other tasks.**
>
> A4: Thanks for pointing out it. The underlying reasons are:
>
> i) Backbone. ViT and CNN work differently.
> ViTs process image patches sequentially and use attention mechanism to capture global information, resulting in the destroy of spatial information yet it is important in 3D pose.
> In contrast, CNN can leverage local receptive fields through convolutions to capture the spatial dependencies, which is beneficial for 3D pose.
> Given that most of existing methods [14,a,b,c] use CNN to extract 2D pose and then use transformer for 3D estimation, there lacks plain ViT-based 3D pose method.
> Thus we conjecture that plain ViT does not work well on 3D pose until now.
>
> ii) Training. CNNs have been well studied in 3D pose task with various training experiences and tricks, while these experiences are lacked of study in ViT.
> Our utilization of [14] with ViT backbone is only a simple and preliminary attempt to validate the possibility of our HAP on 3D pose task, which is not optimal to achieve good performance.
> We will continue to study ViT on 3D pose task in the future.
>
> [a] Exploiting Temporal Contexts with Strided Transformer for 3D Human Pose Estimation, TMM2022.
>
> [b] Keypoint Transformer: Solving Joint Identification in Challenging Hands and Object Interactions for Accurate 3D Pose Estimation, CVPR2022.
>
> [c] P-STMO: Pre-Trained Spatial Temporal Many-to-One Model for 3D Human Pose Estimation, ECCV2022.
>
> **Q5: The Structure Alignment Loss is appealing since it brings contrastive learning in the paper, but given results in Table 2, it seems that it does not bring a significant gain. The improvements are quite marginal and similar to HAP-2 given a certain variance during the fine-tuning stage (only on MSMT17 results are better than HAP-2).**
>
> A5: There might be a slight misunderstanding here. In fact, HAP-2 is the one which uses the structure alignment loss. Compared with HAP-2 vs. HAP-0 and HAP vs. HAP-1, our structure alignment loss results in +1.5\%/+1.3\% improvement on MSMT17, and +0.7\%/+0.7\% improvement on MPII, respectively.
>
> We also appreciate your positive comments on the appealing structure alignment loss.
>
> **Q6: Other human-centric pre-training method such as PATH are proposing zero-shot or results on downstream task with a frozen encoder. Do you have some results following this fine-tuning strategy (i.e. training only the head). It would be great to see how important it is to fine-tune the encoder for each task. Or maybe using adaptor.**
>
> R6: Thanks for your insightful advice.
>
> i) Unfortunately, our HAP fails on the evaluations of training only the head, achieving largey inferior performance than PATH.
> The reason is that HAP is a self-supervised method that pre-trains on the datasets **different** from that in the downstream tasks, while PATH is a supervised learning method using most of the downstream datasets.
>
> ii) Fine-tuning the encoder for each task is important.
> Under the fine-tuning setting, our HAP performs better than PATH on most of the benchmarks.
> Moreover, PATH also points out that fine-tuning is necessary to obtain high performance on in-dataset, out-of-dataset and unseen-task evaluations.
>
> **Q7: Find a different explanation of "views''.**
>
> R7: Thank you for pointing out this. We would like to accept your suggestion and use "masked images'' to replace "views'' in the revised version.

---

> > ### Comment · Reviewer_QQMw · 2023-08-11
> > **Rebuttal**
> >
> > After reading other reviews and rebuttal from authors I am leaning towards acceptance of the paper, I will move to 'weak accept' since I see a clear consensus between the different reviews. Authors answered all questions that I raised during the first stage of the reviewing procedure.
> >
> > The new results such as the 2D-keypoints pretraining baseline and the pretraining on MSCOCO only show that the proposed method is working well as soon as a large-scale training set is deployed for pre-training.
> >
> > Regarding Q4 on 3D pose, there is a recent paper Humans-4D accepted to ICCV'23 showing that ViT can be used for 3D pose estimation. This is just a remark for the author.
> >
> > I am still not really convinced by the Structure Alignement Loss, I would suggest authors to downgrade their claim regarding this 'novelty'.
> >
> > I appreciate answer to Q6; authors clearly explain that fine-tuning the encoder is needed for achieving good results. Using the encoder frozen is failing which is a bit surprising; one comment can be added in the main paper.

---

> > > ### Author Response · Authors · 2023-08-14
> > > **Response to Reviewer QQMw (2)**
> > >
> > > We greatly appreciate your positive and encouraging feedback regarding our work and response.
> > >
> > > i) We concur with your viewpoint that our method ``is working well as soon as a large-scale training set is deployed for pre-training''.
> > >
> > > ii) We really thank you for pointing out a recent paper Humans-4D [a], which is a ViT-based work for the 3D pose estimation.
> > > We will cite and study this work, and further integrate its experiences to our HAP.
> > >
> > > iii) Thanks for your constructive suggestion on the Structure Alignment Loss.
> > > We will carefully revise our paper to downgrade the claim regarding the novelty of the Structure Alignment Loss.
> > >
> > > iv) Thanks for your advice. We will add a comment on the failed results when using the frozen encoder in the main paper following your suggestion.
> > >
> > > We appreciate your insights and suggestions that are valuable for us to improve the quality of our paper.
> > > We will carefully revise and refine our manuscript following your suggestions.
> > > Thanks again!
> > >
> > > [a] Goel, Shubham, Georgios Pavlakos, Jathushan Rajasegaran, Angjoo Kanazawa, and Jitendra Malik. Humans in 4D: Reconstructing and Tracking Humans with Transformers. ICCV2023.

---

### Author Rebuttal · Authors · 2023-08-10

We thank all the reviewers for their insightful and  constructive comments. We are encouraged that reviewers generally recognize the strengths of our paper in:

- Method: positive impact [QQMw], simple and effective [BM1x], make sense [QQMw], well motivated [ueQY], intuitive and appropriate [rHqg], straightforward and promising [rHqg].
- Experiment: sufficient and extensive [BM1x, ueQY], good and informative ablation studies [QQMw, BM1x], good performance [QQMw, ueQY].
- Presentation: clear [QQMw, rHqg], well-written [QQMw, BM1x, ueQY], well-constructed [QQMw, rHqg], well-described details [BM1x].

The point-wise responses have been provided below. We hope our responses can clarify all reviewers' confusion and alleviate all concerns.

Note that due to the limitation of rebuttal time, we only conducted pre-training for 100 epochs for the additional experiments suggested by the reviewers. In the future, we will expand to 400 epochs to further demonstrate the reliability of the conclusion.

---

### Decision · Program_Chairs · 2023-09-21

**Decision:**

Accept (poster)

**Comment:**

This work introduces human structure prior to human-centric pre-training, achieving performance gain on multiple downstream human-centric benchmarks.

Although this paper got a borderline score (averaged 5.25), all reviewers rated positively. During the rebuttal & discussion process, several concerns like baseline settings, more ablation studies, and the effect of occlusions raised by the reviewers are partly addressed by the author.

Although the novelty is somewhat limited (as mentioned by Reviewer QQMw, ueQY, and rHqg), I agree with the reviewers that this work may have a positive impact on the human-centric vision community as it reveals the importance of introducing human prior.

The AC is inclined to accept this paper.